# Palliative Care for Patients with End-Stage, Non-Oncologic Diseases—A Retrospective Study in Three Public Palliative Care Departments in Northern Italy

**DOI:** 10.3390/healthcare10061031

**Published:** 2022-06-02

**Authors:** Massimo Romanò, Sabina Oldani, Valter Reina, Michele Sofia, Claudia Castiglioni

**Affiliations:** 1Organizing Committee—Postgraduate Master in Palliative Care, University of Milan, 20122 Milan, Italy; 2Palliative Care ASST Milano Ovest, 20020 Milan, Italy; sabina.oldani@asst-ovestmi.it (S.O.); claudia.castiglioni@asst-ovestmi.it (C.C.); 3Palliative Care ASST Valle Olona, 21052 Varese, Italy; valter.reina@asst-valleolona.it; 4Palliative Care ASST Rhodense, 20017 Milan, Italy; medico11@hotmail.com

**Keywords:** palliative care, end-of-life care, supportive care, hospice, non-cancer illness

## Abstract

Patients with irreversible malignant and non-malignant diseases have comparable mortality rates, symptom burdens, and quality of life issues; however, non-cancer patients seldom receive palliative care (PC) or receive it late in their disease trajectory. To explore the characteristics of non-cancer patients receiving PC in northern Italy, as well as the features and outcomes of their care, we retrospectively analyzed the charts of all non-cancer patients initiating PC regimens during 2019 in three publicly funded PC departments in Italy’s populous Lombardy region. We recorded the baseline variables (including data collected with the NECPAL CCOMS-ICO-derived questionnaire used since 2018 to evaluate all admissions to the region’s PC network), as well as treatment features (setting and duration) and outcomes (including time and setting of death). Of the 2043 patients admitted in 2019, only 12% (243 patients—131 females; mean age 83.5 years) had non-oncological primary diagnoses (mainly dementia [*n* = 78], heart disease [*n* = 55], and lung disease [*n* = 30]). All 243 had Karnofsky performance statuses ≤ 40% (10–20% in 64%); most (82%) were malnourished, 92% had ≥2 comorbidities, and 61% reported 2–3 severe symptoms (pain, dyspnea, and fatigue). Fifteen withdrew or were discharged from the study PCN; the other 228 remained in the PCN and died in hospice (*n* = 133), at home (*n* = 9), or after family-requested transfer to an emergency department (*n* = 1). Most deaths (172/228, 75%) occurred <3 weeks after PC initiation. These findings indicate that the PCN network we studied cares for few patients with life-limiting non-malignant diseases. Those admitted have advanced-stage illness, heavy symptom burdens, low performance statuses, and poor survival. Additional efforts are needed to improve PCN accessibility for non-cancer patients.

## 1. Introduction

Palliative care (PC), according to the World Health Organization (WHO), should be made available to all patients with special needs resulting from advanced, life-threatening diseases, including, but not limited to, cancer [1]. Each year, throughout the world, an estimated 40 million people (mainly those living in less developed countries) require PC, but fewer than 15% receive it [1]. WHO data from 2014 [2] revealed diagnoses of cancer in only 34% of the adults with documented PC needs. In the vast majority of cases, the primary diagnosis was non-oncological; in most cases, it was chronic cardiovascular disease (38.5%), chronic lung disease (10.3%), HIV/AIDS (5.7%), diabetes (4.5%), chronic kidney disease (CKD) (2%), liver cirrhosis (1.7%), and Alzheimer’s disease or other dementias (1.6%). This last category of diseases is already growing in importance: dementias are now expected to be the non-cancer conditions that will have the greatest impact on patients’ quality of life over the next 40 years [3].

In light of the above considerations, the European Association for Palliative Care published a white paper in 2009 with recommendations for implementing PC in Europe, not only for patients with malignancies but also those with advanced, chronic, non-oncological diseases [4]. The latter diseases are, in fact, associated with substantial symptom burdens, i.e., physical (including, but by no means limited to, pain), psychological, and spiritual [5], and patients suffering from these conditions have twice as many PC needs as those with terminal cancer [2,6]. 

However, substantial clinical and epidemiologic differences have been documented between patients receiving PC for non-malignant vs. malignant disease. The cancer patients tend to be younger, male, and with better functional statuses, and they are generally admitted to PC programs earlier, whereas those with chronic non-oncological diseases (dementia, stroke, and heart failure) have poorer prognoses (<1 month) and low Palliative performance statuses [7] (10–20% [8]).

Published evidence on the benefits of PC is still based largely on its use in patients with cancer [5]. However, recent data show that, individuals with terminal non-cancer diseases (heart failure and other forms of organ failure, chronic obstructive pulmonary disease (COPD), and CKD) who receive PC during the last six months of their lives have lower frequencies of emergency department visits, hospitalizations, and admissions to intensive care units than their counterparts who do not receive PC [9]. 

In recent years, several assessment tools have been developed to facilitate the early identification of all patients with PC needs (the Gold Standard Framework, Prognostic Indicator Guidance, Supportive and Palliative Care Indicators tool, and NECPAL CCOMS-ICO tool) [10,11,12]; however, even with these supports, patients with non-oncologic conditions continue to be under-represented among those receiving PC [8]. A NECPAL CCOMS-ICO-derived tool is currently being used in the Lombardy region of Italy to identify patients with actual PC needs [13]. Lombardy is Italy’s most populous region, with a total of 10,103,000 residents in 2019. It is also the region with the most highly developed publicly funded PC network, which includes 73 hospices and 131 home-based care units, providing care for 29,900 patients in 2019. In the study described below, we retrospectively investigated a cohort of patients with advanced non-oncological diseases who were cared for through the dedicated PC facilities of three of Lombardy’s local health departments, which include four hospices. Our aims were to characterize the baseline profile of this cohort, as well as the features and outcomes of their care. 

## 2. Materials and Methods

A retrospective cohort study was conducted in three large publicly funded healthcare departments (*Azienda Socio Sanitaria Territoriale*, ASSTs) serving extra-urban populations in Lombardy: the Rhodense ASST, which has a catchment population of approximately 485,000; the Valle Olona ASST (catchment population: ~ 430,000); and the ASST of Western Milan Province (catchment population: ~ 470,000). Each ASST has a dedicated PC department that provides intra-/extra-hospital consultation services and delivers palliative care and pain therapy in diverse settings, including four separate hospices, with a total of 44 beds, home-care units, outpatient clinics, and, in the case of Rhodense ASST and Milano Ovest ASST, day-hospital and -hospice units. Applications/referrals for care in all these three PC departments (referred to hereafter as the study PC network or PCN) are assessed by a PC specialist, who interviews the patient and/or family and verifies the patient’s actual PC needs, with the aid of the NECPAL CCOMS-ICO-derived tool [12,13]. 

We retrospectively reviewed the charts of all patients who were consecutively admitted to PC network in 2019. The baseline data recorded included: patient demographics; origin of PC referral (primary care physician; hospital staff; nursing-home staff); primary diagnosis (as established by the referring PC physician); symptoms (presence/absence of dyspnea, pain, fatigue); the Karnofsky performance status (KPS) (scores ranging from 0 to 100, with higher scores indicative of a greater functional capacity and better prognosis) [14]; and the clinical indicators of disease severity/progression defined in the NECPAL CCOMS-ICO checklist [12,13]. The latter included both general indicators (hospitalizations during the past 12 months, co-morbidities, and presence/absence of malnutrition [15], as well as those that were specific to the primary diagnosis). We also recorded the palliative care characteristics (delivery settings (i.e., home and hospice), unplanned transfers to an acute-care facility), and outcomes (discharge to another healthcare facility, voluntary withdrawal from the study PCN, and in-network mortality (time and setting of death)). 

Time of death was classified in accordance with the prognostic classes defined by the Palliative Prognostic Index: <3 weeks, 3–6 weeks, and >6 weeks [16].

We have received the approval of the study protocol from the Healthcare Directions of all ASST. These formal approvals are available from the corresponding author. 

## 3. Results

### 3.1. Baseline Patient Profiles

In the year 2019, a total of 2043 patients were enrolled in the palliative care programs administered by one of the three ASSTs making up the study PCN. Our study cohort comprised the 243 (12%) patients suffering from chronic non-oncological diseases. Table 1 shows their demographic and clinical characteristics at the time of PCN admission. The two most common primary diagnoses were dementia (*n* = 78, 32%) and chronic heart disease (*n* = 55, 23%). In roughly two-thirds (64%) of the 243 cases, the PCN referral was made during a hospital admission (general medicine wards [39%] and specialty wards [20%]). In the remaining cases, the referral was made by general practitioners caring for patients in the latters’ homes (35%) or less commonly in a nursing home (1%). 

As for the general indicators of disease severity/progression defined in the NECPAL tool (Table 1), over half (53%) of the 243 patients had histories of ≥2 unscheduled hospitalizations during the year preceding PCN admission; 199 (82%) patients were malnourished; the majority were suffering from fatigue (63%), pain (55%), and/or dyspnea (53%); and 224 (92%) had two or more comorbidities (detailed in Figure 1).

All 243 had KPS scores of ≤40%, and two-thirds of the scores were 10–20%. In terms of the disease-specific indicators of severe/progressive disease listed in the NECPAL-derived tool, all 243 patients met the minimum requirement for PC eligibility.

### 3.2. Palliative Care Characteristics and Outcomes

In 141 (58%) of the 243 cases, the palliative care was delivered entirely in a hospice setting. Eighty-five other patients (35%) were cared for exclusively in their homes (Table 2), and seventeen (7%) were cared for in both settings. 

In 15 (6%) of the 243 cases, the care being delivered by the PC network was interrupted, and the patient was discharged. In three of these cases, the decision to terminate PC was made by the patient or their family, and no specific reasons were given. The remaining 12 discharges involved five patients who were referred for non-palliative care at home, two who were referred to the care of their primary-care physicians, one who was referred for re-evaluation by a cardiologist, and four who were transferred to another residential/inpatient healthcare facility, i.e., a nursing home (*n* = 1), another hospice (*n* = 1), an acute-care hospital (*n* = 1), or a rehabilitation facility (*n* = 1).

The other 228 (93.8%) of the patients died while still enrolled in the study PCN (Table 2), and in 175 (76.8%) of these cases, the death occurred within 3 weeks of PC admission. A total of 133 (58%) of the 228 deaths occurred in hospice, 94 (42%) occurred in the patient’s home, and in the remaining cases, death occurred shortly after the patient had been transferred to the emergency department at the family’s request. 

A total of 172 patients (75% of decedents) died within 3 weeks after the enrollment in the PCN.

## 4. Discussion

Our study represents the first attempt to explore the profiles and clinical pathways of patients with advanced chronic non-oncological diseases and enrolled in home- and hospice-based PC programs administered by regional public healthcare facilities in the Lombardy region of Italy. In the year 2019, a total of 2043 patients initiated care within the study PCN (total catchment population: 1,385,000). The vast majority of these patients were suffering from cancer: only 12%—the 243 patients we investigated—had primary diagnoses that were non-oncological. 

These findings are consistent with findings from a previous study in Italy, which found that patients with non-oncologic diseases who were receiving care through a publicly funded PC network accounted for only 5% of the home-care services delivered [17]. A similar picture emerged from the DEMETRA study, an observational study conducted in five Italian regions in home care and hospice settings. Of the 1013 patients enrolled in this study over the course of 18 months, only 148 (14.6%) had non-oncological diagnoses: 3.5% of the patients in this cohort had cardiovascular disease, 2.6% had dementia, and 2.5% were suffering from chronic lung disease [18].

In contrast, the data reported for 2019 in the United States by the National Hospice and Palliative Care Organization showed that more Medicare hospice patients had a principal diagnosis of Alzheimer’s disease/dementia/Parkinson’s disease than any other disease. Principal diagnoses categories of stroke, respiratory disease, and circulatory/heart disease have grown the most since 2014 [19]. In a British primary-care setting, the greatest increase in accesses to palliative care from 2009 to 2014 involved dementia (from 20.9% to 40.7% of all cases), whereas smaller increases were seen in the percentages of accesses by patients with heart failure (from 12.6% to 21.2%) and COPD (from 13.6% to 21.2%). On the whole, however, there was still a clear predominance of cancer patients in this setting (increased from 57.6% to 61.9%) [20]. 

Our cohort was characterized by advanced age (mean: 83.5 years) at admission, and this feature was particularly striking in the female patients (85.4 years), who accounted for over half of the cohort members. All 243 patients had Karnofsky performance status scores of ≤40%, one-fifth were already experiencing pain, dyspnea, and fatigue, and three-quarters survived less than 3 weeks. Over 82% were also malnourished, a finding that not only reflects severe/progressive disease but also suggests that nutritional issues have received insufficient attention in the earlier stages of the disease [21]. 

These findings indicate that, as in other countries [22,23,24], in the area of Lombardy under study, patients with life-limiting chronic diseases other than cancer are being “intercepted” by the public health system’s PC network when their diseases are far advanced, clinically complex, and burdened by multiple symptoms that impact quality of life, in the same ways as cancer patients (5). This was especially true of patients with chronic heart disease: 98% had >2 comorbidities, 56% had been hospitalized >2 times during the last 6 months, almost 90% reported 2 or 3 symptoms, and three-quarters were experiencing significant pain. Substantial differences between the proportion of heart-failure patients with PC needs and those who actually receive PC are well-documented, as is the tendency to postpone the initiation of PC in these patients [24,25,26]. 

Consistent with the above findings, almost two-thirds of our non-cancer patients (64%) had been referred for PC during a hospital stay, and over half of these referrals came from acute-care wards [27]. This finding suggests that: (1) these patients are likely to receive potentially inappropriate aggressive treatments, even during the most advanced stages of their disease, when cures are extremely unlikely, and (2) PC tends to be reserved exclusively for the end of life, an intervention regarded by some (healthcare providers and patients) as a “grim-reaper service” [28]. 

Early initiation of palliative care can be hindered by an insufficiently large work force of physicians who are specialized in this field. In the United States, the ratio of palliative care specialists to patients enrolled in palliative care programs for the year 2018 was 1:808, and the situation is expected to worsen by 30%, owing to physician burnout and aging/retirement [29]. Other barriers to the early initiation of palliative care are particularly important when patients have chronic diseases other than cancer [30], such as the increased difficulties involved in formulating a short-to-medium term prognosis and identifying the terminal phase of such diseases. Some authors feel that the term “palliative care” itself is also an obstacle [30] because it is identified by many as an intervention reserved solely for the end-of-life phase, when hope has vanished and all efforts to ameliorate the underlying disease will be suspended. The stigma associated with the term “palliative” [31] is encountered among patients and their families but also among their physicians, particularly those in non-oncological branches of medicine, who may be less accustomed to discussing prognosis and end-of-life issues with patients and their families than oncologists [32]. 

The limitations of our study include its retrospective nature and the possibility that the data we collected are incomplete. However, this risk is minimized by the fact that regional regulations require that all patients admitted to the study PCN be evaluated with the same assessment tool [13]; therefore, it is therefore unlikely to affect the significance of our results. Another important limitation is the absence of the data regarding the analysis of quality of life and symptom relief in our cohort.

This study was designed in the last months of 2020 and analyzed the data regarding 2019, before the COVID-19 pandemic period. The pandemic has increased the number of patients admitted to home care and reduced those admitted to hospice setting. 

In the summer of 2021, the situation normalized, with a stable increased number of patients admitted to home care. 

## 5. Conclusions

In conclusion, our study revealed a significant delay in the initiation of palliative care in patients with advanced, life-limiting nononcologic diseases, despite the WHO recommendations in this regard [1], and very low survival rate after PCN admission. Further efforts should be made to improve and facilitate accessibility to PCN for this important patient population. 

## Figures and Tables

**Figure 1 healthcare-10-01031-f001:**
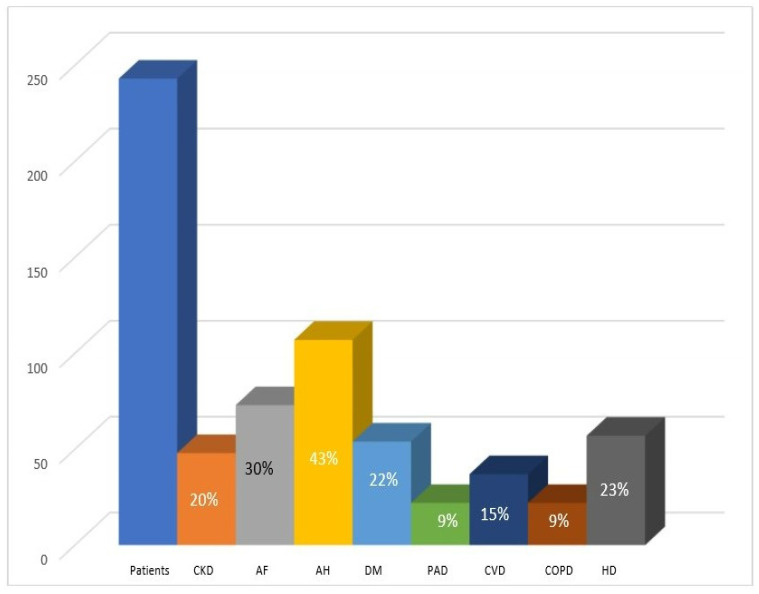
Distribution of co-morbidities by primary diagnosis. CKD: chronic kidney disease; AF: atrial fibrillation; AH: arterial hypertension; DM: diabetes mellitus; PAD: peripheral artery disease; CVD: cerebrovascular disease; COPD: chronic obstructive pulmonary disease; HD: heart disease.

**Table 1 healthcare-10-01031-t001:** Clinical characteristics of the patient cohort at PCN admission.

	Total Cohort	Primary Diagnosis						
		Dementia	HD	Lung Disease	Stroke	Neurological Disease	CKD	LD
**Patients (*n*/%)**	243 (100%)	78 (32%)	55 (23%)	30 (12%)	29 (12%)	22 (9%)	17 (7%)	12 (5%)
**Males (*n*/%)**	112 (46.1%)	24 (31%)	21 (38%)	17 (57%)	16 (55%)	16 (73%)	11 (65%)	7 (58%)
Age (mean − years)	83.5	85.7	85.7	81.8	84.6	76.6	83.4	74.5
**Females (*n*/%)**	131 (53.9%)	54 (69%)	34 (62%)	13 (43%)	13 (45%)	6 (27%)	6 (35%)	5 (42%)
Age (mean − years)	85.4	87.4	86.7	82.8	84	78.1	86.3	75.2
**Origin of PCN referral**								
Acute-care hospital	156 (64%)	43 (55%)	35 (64%)	24 (80%)	23 (79%)	10 (45%)	14 (82%)	7 (58%)
Home	84 (35%)	33 (42%)	20 (36%)	6 (20%)	5 (18%)	12 (55%)	3 (18%)	5 (42%)
Nursing home	3 (1%)	2 (3%)			1 (3%)			
**Baseline clinical findings**								
General indicators of disease severity ^a^								
History of ≥2 urgent hospitalizations ^b^	128 (53%)	34 (44%)	31 (56%)	22 (73%)	13 (45%)	9 (39%)	8 (47%)	11 (92%)
≥2 comorbidities	224 (92%)	70 (89.7%)	54 (98%)	30 (100%)	28 (97%)	17 (78%)	16 (94%)	9 (75%)
Malnutrition ^c^	199 (82%)	70 (90%)	41 (74%)	21 (70%)	24 (83%)	16 (74%)	15 (88%)	12 (100%)
All 3 of the above	101 (42%)	30 (39%)	22 (40	15 (50%)	12 (41%)	7 (30%)	7 (41%)	8 (67%)
KPSS 20–10% 30–40%	155 (64%) 88 (36%)	60 (77%) 18 (23%)	26 (48%) 29 (52%)	12 (40%) 18 (60%)	22 (75%) 7 (25%)	15 (68%%) 7 (32%)	12 (71%) 5 (29%)	8 (64%) 4 (36%)
**Most common symptoms**								
Pain	133 (55%)	39 (50%)	41 (73%)	13 (44%)	19 (66%)	10 (43%)	9 (54%)	2 (17%)
Dyspnea	128 (53%)	30 (39%)	39 (69%)	29 (95%)	13 (42%)	9 (41%)	8 (47%)	0
Fatigue	153 (63%)	38 (49%)	48 (86%)	21 (68%)	12 (41%)	18 (78%)	12 (70%)	4 (33%)
≥2 of the above	148 (61%)	32 (42%)	49 (89%)	23 (77%)	16 (55%)	12 (56%)	10 (59%)	6 (50%)
None of the above	12 (5%)	6 (8%)	(0%)	(0%)	4 (14%)	2 (9%)	(0%)	(0%)

^a^ As defined by the NECPAL CCOMS-ICO tool (ref.). ^b^ During the 12 months preceding PCN enrollment. ^c^ Abbreviations: KPSS, Karnofsky performance status scale; PCN, palliative care network. Reflected by ≥1 of the following during the 6 months preceding enrollment: serum albumin <2.5 g/dL, >10% decrease in body weight, clinical perception of persistent, intense/severe, progressive, and irreversible nutritional deterioration unrelated to intercurrent conditions, HD: heart diseases, CKD: chronic kidney diseases, LD: liver diseases.

**Table 2 healthcare-10-01031-t002:** Features and outcomes of palliative care ^a^.

	Total Cohort	Primary Diagnoses						
		Dementia	HD	Lung Disease	Stroke	Neurological Disease	CKD	LD
**Patients (*n*/%)**	243 (100%)	78 (32%)	55 (23%)	30 (12%)	29 (12%)	22 (9%)	17 (7%)	12 (5%)
**Palliative care setting** ** Hospice (*n*/%)** ** Home (*n*/%)** ** Both (*n*/%)**	141 (58%) 85 (35%) 17 (7%)	44 (56%) 34 (44%)	28 (51%) 27 (49%)	19 (63%) 11 (37%)	23(79%) 6 (21%)	10 (48%) 12 (52%)	14 (82%) 3 (18%)	7 (58%) 5 (42%)
**In-network mortality**	228 (93.8%)	72 (92%)	49 (89%)	29 (97%)	29 (100%)	20 (91%)	17 (100%)	12 (100%)
**Time of death ^b^** ** <3 weeks** ** 3–6 weeks** ** >6 weeks**	172 (75%) 21 (9%) 35 (16%)	53 (74%) 6 (8%) 13 (18%)	34 (69%) 5 (10%) 10 (21%)	21 (73%) 3 (10%) 5 (17%)	24(83%) 2 (7%) 3 (10%)	12 (60%) 4 (20%) 4 (20%)	17 (100%)	11 (92%) 1 (8%)
**Setting of death** ** Hospice** ** Home** ** Acute care**	133 (58.4%) 94 (41.2%) 1 (0.4%)	30 (42%) 42 (58%) 0	26 (53%) 22 (45%) 1 (2%)	19 (66%) 10 (34) 0	24 (83%) 5 (17 0	13 (68%) 7 (32%) 0	14 (82%) 3 (18%) 0	7 (58%) 5 (42%) 0

Abbreviations: PCN, palliative care network, HD: heart diseases, CKD: chronic kidney diseases, LD: liver diseases. ^a^ All results are presented as *n* (%), unless otherwise stated. ^b^ From admission to study PCN.

## Data Availability

The data presented in this study are available on request from the corresponding author. The data are not publicly available due to privacy reasons.

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
