# Peer review of "Palliative Care for Patients with End-Stage, Non-Oncologic Diseases—A Retrospective Study in Three Public Palliative Care Departments in Northern Italy"

_healthcare, 2022, doi:10.3390/healthcare10061031_

Round 1
Reviewer 1 Report
- The introduction and the problem statement are well addressed.
- At lines 86-90: you mention how one department of your study addresses palliative care referrals. However, how do the other two departments work in this regard?
- At the Methodology section it is missing several kinds of information: i) the type of sampling technique used; ii) the ethics procedures performed and approvals received before the start of the study; iii) the type of statistical data to be performed should be described.
- Quality of the Tables and Figures should be improved.
- Table 1: data in frequencies and percentages are not always consistently presented and makes confusion for the reader. For instance, this happens in the females' line. Also, there is some information in the table note highlighted in yellow.
- Figure 1 should be presented more clearly. The names and percentages of the comorbidities are very close.
- For the discussion section:
- Are there differences between females and males? If so, what clinical implications this result may imply?
- You mention that your results are consistent with others also performed in Italy. But, do you know if there are differences in data from other regions where the palliative care services are not so developed as in Lombardy?
- You mention that health care staff are in need of more training or knowledge to refer patients to palliative care services. This should be specifically addressed and more developed for these professionals dealing with non-cancer patients and from an early or timely approach. The literature in this line is not so much developed but there are a few papers published during the last years that should be studied.
- On the basis of your results and from previous ones, are there differences between the symptomatology or needs of patients with non-cancer diseases vs. cancer?
- I consider that you are making a very strong conclusion saying that patients with non-oncological diseases receive a later referral to palliative care in comparison to those with cancer. However, in your study, you are not considering cancer patients, so these conclusions are not based on your evidence. This should be amended accordingly to your type of study ad main purpose.
- Ho do you consider that COVID-19 has impacted the profile of this type of patient and in their referral to palliative care?
Author Response
Comment 2: There was an error in translation. We deleted "one of", because the patients are referred to all PC units in the same manner.
Comment 3, i) All patients admitted to PC network during the 2019 are the object of the study. We changed the text as follows: "The charts of all patients, consecutively admitted to PC programs.
ii) We received the approvals of the study protocol from the healthcare directions of all ASST. The documents are available from the corresponding Author.
iii) This study did not include statistical analysis
Comment 4. We have improved the tables and the figure
Comment 5: We have modified the table 1, according to the reviewer's suggestions
Comment 6: We modified the figure, according to the reviewer'suggestions.
Comment 7 (Discussion): The only differences between females and males are the mean age.
The two Italian studies we mentioned in our discussion have collected data from different settings in different areas: so it is impossible to compare those data with our data, collected in amore homogenous areas. Moreover the Demetra study Authors stated that many patients were lost to follow-up, introducing a bias.
We did mentioned that healthcare staff is in need of more training, but the number of palliativists is insufficient in relation to that of patients.
We have added that the patients with non cancer diseases have the same symptom burden of cvancer patients.
We made a conclusion that the patients with non cancer illnesses receive a late referral to PC, but we did not mentioned a comparison with cancer patients, non included in our analysis, as outlined in the text. We stated that" patients with life-limiting chronic diseases other than cancer..."
Lastly we have no data regarding the impact of COVID 19 on the profile of this kind of patients.
Reviewer 2 Report
Very interesting article, however there are some shortcomings that the authors should work on. Firstly, the Materials and Methods should be extended. In this section, the authors should add confirmation in the literature of the method used.
Secondly, the quality of Table 1 leaves much to be desired. It is suggested to post it in an edited version and not as a photo. That way the quality will be satisfactory.
Thirdly, the Figures 1 and 2 need some improvement in order to be more readable.
The Article does not have any conclusions. This sections must be added.
Author Response
We have modified the tables and figures, according to the Reviewer's suggestions.
Regarding the confirmation in literature of the method used, there are no references, to our knowledge, because the charts we have analyzed are collected following a specific, Lombardy region defined model.
Reviewer 3 Report
Dear authors
It is a valuable article, and it is one of the topics that has been under-discussed. In general, after the general correction, which includes completing the introduction and Also, the discussion and conclusion section of this article will be ready to be published. Moderate changes are required to the English language and style
Author Response
We thank very much the Reviewer, but we did not understand what he means, when he writes: "completing the introduction and Also (we suppose Materials and Methods). We would like to know how we may improve our paper, according to the Reviewer.
Regarding the English changes required, our paper was written and rechecked by a native English-speaking colleague, as suggested by Assistant Editor.
Round 2
Reviewer 1 Report
Dear authors,
Thanks for addressing the comments highlighted in my first review.
I detect that the Comment 2 is marked in the text as a comment, but not solved.
My suggestions for the Discussion section were aimed to extend and improve your reflection and analysis. This is the case of the COVID-related one; I am aware that this is not your scope. However, I was wondering how your results could be useful or interesting to be taken into consideration in a health context where COVID has changed everything and, especially, for those patients that are especially vulnerable.
Author Response
Dear Reviewer, I have addressed your last suggestions, modifying the text at 86 line and adding a brief comment about Covid 19 at the end of discussion.